# A Hybrid Approach to Reassemble Ancient Decorated Block Fragments through a 3D Puzzling Engine

**Roberto de Lima-Hernandez *** and **Maarten Vergauwen**

Department of Civil Engineering, TC Construction—Geomatics, KU Leuven—Faculty of Engineering Technology, 9000 Ghent, Belgium; maarten.vergauwen@kuleuven.be

*   Correspondence: roberto.delimahernandez@kuleuven.be

**Abstract:** The reassembling of severely damaged tangible heritage is a primordial task for archaeologists who not only aim to further study the past but also to preserve ruined ancient monuments. As a consequence, various researchers have proposed methods to automatically solve this problem by computing and matching geometric properties of counterpart fragments. Although their results are quite promising, experts still carry out this task manually by finding relationships between distinctive matching cues, such as type of decoration, remaining traces, inscriptions' content, etc. The topic itself poses challenges to both automatic and manual approaches due to the high level of damage ancient broken fragments have undergone over the centuries. Therefore, this paper proposes a Puzzling Engine that combines crucial elements of automatic and manual methodologies to empower experts with registration tools for reassembling fragmented heritage. Unlike similar hybrid human-computer puzzling engines, our approach is capable of automatically proposing matches and rough alignments solely based on the geometry of fractured surfaces. Based on these initial solutions and a set of registration tools, experts can accurately solve the puzzle. The virtual environment has been used and verified to find pairwise puzzle-pieces of actual antique wall decorated fragments, resulting in new discoveries that experts could not have come up with by utilizing classic techniques. Concretely, the contributions are twofold, (i) a feature-based registration pipeline that is able to suggest both matches and alignments to the user and (ii) a virtual interface that integrates automatic and user-assisted techniques to accurately puzzle fragmented surfaces.

**Keywords:** heritage reassembling; local 3D descriptors; feature matching

## 1. Introduction

Many ancient heritage sites have undergone numerous transformations over time to such a degree that their archaeological remains have become fragmented. The incompleteness of such monuments and artifacts prevents experts from further studying them. Consequently, the reassembling of fragments is a paramount task not only for research purposes but also for heritage conservation. The broken pieces are typically reassembled based on several factors, such as the place where the fragments were found, the type of decoration, inscriptions content, etc. The task of identifying such matching cues is cumbersome due to the severe damage ancient artifacts have suffered. As a result, archaeological research is delayed since the duration of field digging campaigns is typically rather limited, as is the time allowed in museums or archives to study the assets. Furthermore, manual manipulation might jeopardize the objects' physical and archaeological integrity. Therefore, experts have resorted to 3D recording technologies to scan the fragments on-site and digitally reassemble them later on in the laboratory. To this end, various digital platforms have been developed to provide restorers with the tools for visualization in 3D space, as well as with tools for seamless alignment [1]. Although these approaches highly benefit the restoration process, digital puzzling requires specific

expertise not only to retrieve potential matching relationships but also to interact with the recorded models. This paper presents a Puzzling Engine that allows both experts and laymen without a specific background to seamlessly interact with the recorded fragments by means of a set of tools for visualization and registration.

This work is an extension of the Puzzling Engine introduced in [2]. We received an invitation from the organizers of the International Committee of Architectural Photogrammetry (CIPA) symposium to extend the conference paper, which describes a virtual interface that enables experts to reassemble multiple wall broken fragments. Its main characteristics are threefold: (i) The 3D models interaction along a 2D plane, (ii) the contour segmentation that allows to intuitively find common indentation regions between counterpart fragments, and (iii) the registration tools to accurately solve the puzzle. In this extended version, the contour segmentation is improved by incorporating a machine-learning module to extract decorated surfaces. Moreover, computational modules for automatic matching are included so that the engine itself is able to propose potential correspondences between similar fragments. An overview of the computational elements that compose the proposed approach is depicted in Figure 1. The steps presented in our workflow are adapted from the common pipeline for automatic 3D registration, including segmentation, descriptors extraction, feature matching, and object alignment. The segmentation process mainly aims at delineating the footprint of fractured regions by extracting the area around the main decorated plane. Based on the obtained point-cloud, local features are computed to accurately describe the contour surface in terms of its geometric properties. Along with these features, geometric constraints such as normal coherence are taken into account to estimate a set of consistent correspondences. These matched points serve in two ways: As a visual aid to detect possible joint regions and as a basis to compute a rigid transformation for automatic solving. The proposed pipeline is tested over numerous ancient decorated wall fragments of ancient Egyptian tombs to showcase the potential of the Puzzling Engine. Although the obtaining matching results show that for some pair of fragments the registration pipeline estimates accurate alignments, this approach is not meant to substitute automatic objects reassembling methods. The goal is to complement manual alignment tools with geometry-based matching algorithms to ease the puzzling of tangible heritage.

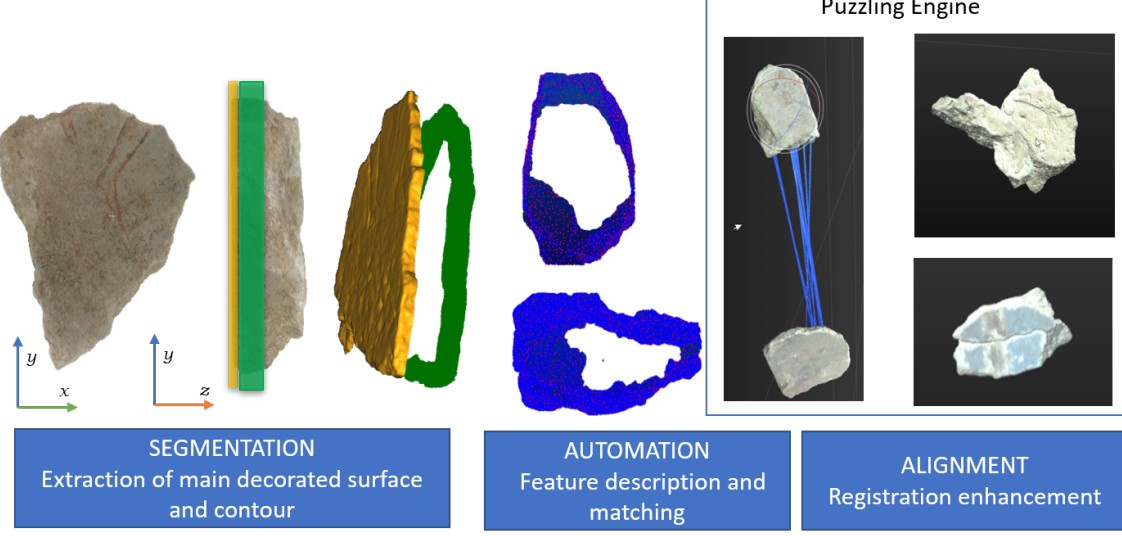

**Figure 1.** Overview of the proposed registration pipeline for pairwise puzzling. The Segmentation stage aims at extracting both the main decorated surface and contour of wall decorated fragments. These regions of interest are processed in the Automation stage, which consists of a feature-based registration pipeline to propose alignment solutions. The final stage is the virtual interface that allows the user to finely align matching points between counterpart fragments.

The key contributions of the presented work are the following.

- Extraction of highly damaged wall decorated surfaces by combining machine learning methods and plane-fitting;
- Normal Coherence-based adaptations of well-known 3D descriptors and feature correspondence algorithms to tackle the automatic reassembling of digital heritage.
- Accurate user-assisted registration of wall broken fragments by means of projecting the input assets into a generic plane so that both objects' maneuvering and alignment transformations can be carried out in 2D space.

The rest of the paper is organized as follows. Section 2 gives an overview of the existing approaches for fragments reassembling. Section 3 describes in detail the proposed methodology from the approach to extract the main decorated surface and fragment's contour, to the automatic alignment. The main core of the paper's contribution is presented in Section 3.2, in which every step of the automatic alignment workflow is explained. The procedure to compute the ground truth metrics and performance assessment of the proposed computational modules are presented in Section 4. Finally, the conclusion and future work are given in Section 5.

## 2. Related Work

With the advent of robust 3D recording technologies and techniques, novel approaches for heritage reassembling have emerged. The accurate geometry of the obtained models has allowed the expansion of algorithmic possibilities to solve cumbersome manual labor. The existing works for archaeological artifacts reassembling in the three-dimensional space are divided into two branches: Computer-assisted and fully-automated approaches. The former aim at providing experts with digital tools to rapidly find matching cues. The latter rely on feature-based correspondence algorithms to automatically register the broken assets. As the proposed work combines components of both, the literature study outlines the significant works of both branches and their differences with the proposed approach.

Computer-assisted reassembling methods support the manual alignment labor by means of a virtual interface, through which users can maneuver the recorded assets in 3D space while identifying potential matching surfaces. To accurately perform this task, researchers have proposed to segment the input data into multiple regions of interest to serve as a basis to easily distinguish matching points. These segmentation algorithms seek to extract geometry-based properties such as curves, normal surfaces, carvings, and so on. For example, Benedict Brown et al. [3] developed a virtual environment that is capable of finding shape-based pairwise matches of fresco fragments. These preliminary solutions are valuable information for restorers to filter out geometrically incoherent puzzles. Along the same lines, Mellado N. et al. [4] deployed a virtual reality environment to align broken archaeological objects by computing a set of registration methods and constraints such as the Iterative Closest Points (ICP) [5] algorithm, vertex distance, and normal coherence. Additionally, their system takes into consideration the user's feedback to enhance registration accuracy, thus functioning as a real-time interactive registration platform. More recently, Papaioannou Georgios et al. [6] proposed a comprehensive pipeline to reassemble 3D archaeological artifacts including ceramic vessels and broken stone fragments of various shapes and sizes. The developed virtual environment includes a set of segmentation methods as well as registration tools for either pairwise or multi-part matching. These methods encompass fracture classification, region growing segmentation, curved-based features extraction, and fragment penetration penalization methods for fine alignments.

On the other hand, automatic methods seek to find matching fragments by computing and comparing high level geometrical properties. This is a challenging task since fragmented archaeological assets have different characteristics so that specific matching cues are considered for puzzling. For instance, poly-chrome ceramic broken fragments are generally composed of well-defined curves, thus making shape-based matching heuristics along with color comparisons suitable approaches to

find potential joint regions [7]. Similarly, the contour and thickness of pottery fragments are essential features for the automatic reassembling of vessels [8]. Stone fractured fragments, however, are more complex artifacts to deal with due to their erratic shape and erosive material. Therefore, researchers have resorted to sophisticated geometric computations to numerically describe and match regions of interest. The method proposed by Huan Qi-Xin [9] relies on discrete geometric operations [10] computed over salient areas to estimate geometrical relationships among multiple fragments. The set of obtained correspondences is processed by the Random Sample Consensus RANSAC method [11] to remove false positives. Lastly, an accurate rigid body transformation is calculated from the resulting matches, thus completing the automatic alignment workflow. Following the same registration process, Zhang Kang et al. [12] proposed to estimate a matching template from the intact object's surfaces based on the Signature of Histograms of Orientations SHOT [13] descriptor and curvature-based features. Then, potential matching relationships are computed by comparing the initial template to the geometrical information of the fractured pieces. More recently, Qunhui Li et al. [14] presented a pairwise matching approach that compares concave-convex patches extracted along boundary contour surfaces. Afterwards, similar to the aforementioned approaches, the matched areas are accurately aligned by means of the Iterative Closest Points algorithm.

The outlined automatic fragments reassembling methods have been successfully applied to puzzle objects with intact broken surfaces. Consequently, the geometrical overlap between fragments is sufficient for feature-based matching methods to find a vast number of correspondences. This property, however, does not hold for archaeological fragments due to the fact that their fractured surfaces have suffered significant damage, preventing automatic methods from computing reliable correspondences. Therefore, the proposed Puzzling Engine is a hybrid approach that combines a set of registration tools with automatic matching algorithms to ease digital puzzling. More specifically, similar to computer-assisted methods, segmentation algorithms are implemented to extract regions of interest such as boundary contours, curve-based local salient points, and the main planar decorated surface. Moreover, inspired by automatic approaches, a set of matches automatically computed by the engine based on the input models' geometric properties, serves as starting point for user decisions. These matching cues play a fundamental role in the puzzling process as they serve as either a confirmation or starting point for restorers to propose alignment solutions. By exploiting the best properties of such approaches, the proposed engine is capable of accurately puzzling heavily damaged ancient heritage fragments.

## 3. Methodology

In this section we describe the algorithmic steps that composed the Puzzling Engine. These include the extraction of regions of interest and the registration workflow to increase the degree of automation of the virtual platform.

### 3.1. Segmentation of Main Decorated Surface

RANSAC is a well-proven method that fits parametric models to an arbitrary set of noisy observations. In our previous work, this algorithm is deployed to extract the main decorated surface by fitting a plane to the fragment's model. For 93% of the dataset the surface of interest is correctly extracted since the models contained a well-defined plane. However, when testing the algorithm for highly-damaged block fragments, the success rate notably decreases. As noted in Figure 2, multiple planes can be fitted to the fractured fragment but none of them correspond to the actual decorated surface. This problem has to do with the fact that only 3D geometry is considered, leaving aside important characteristics of heritage assets such as color and relief. In fact, these features are critical for experts to identify the fragment's main surface and start the puzzling process. Therefore, the decoration color is taken into account as an extra constraint to consider an extracted plane as a decorated surface.

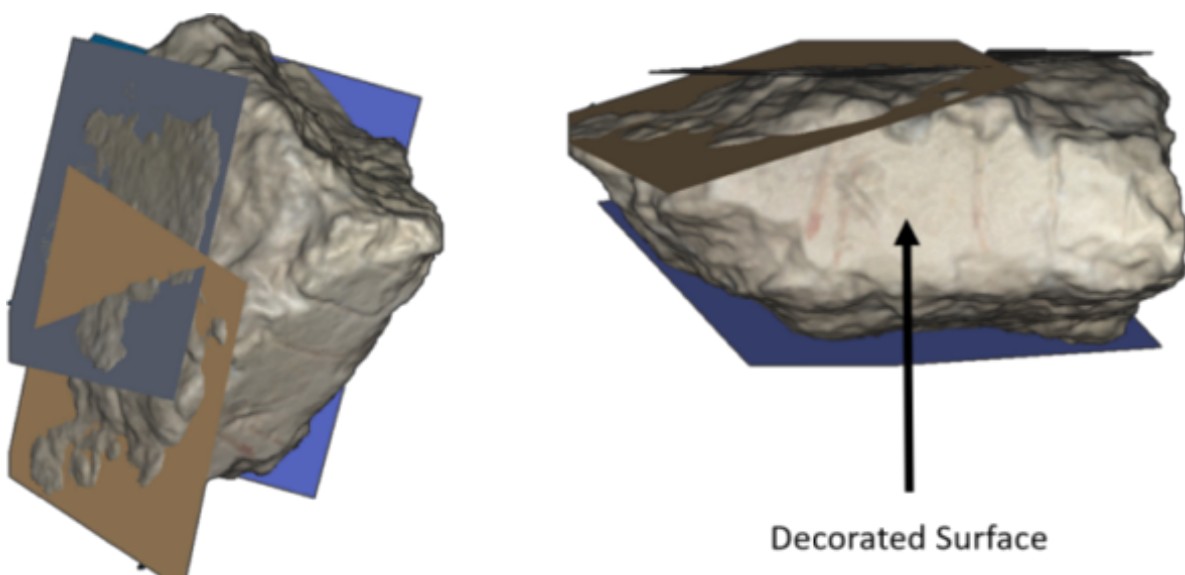

**Figure 2.** Example of a case where RANSAC struggles to find the decorated surface due to the irregular shape of the fragment.

This feature requires us to modify the original approach since we now introduce two additional requirements: We must extract multiple dominant planar surfaces and also need a system capable of scoring the amount of paint on the fragment. These modifications are included in Algorithm 1, which outlines the step-by-step process to segment the main decorated surface. First, the fragment's dominant planes are computed by RANSAC based on a threshold condition. Afterward, for each extracted entity, the percentage of paint or decoration is estimated by a color classifier. Finally, the extracted plane with the highest paint percentage is designated as the main decorated surface. The combination of both characteristics (color and planarity) highly reduces the number of false positives as multiple potential extracted surfaces are evaluated. Since the success of the proposed approach highly relies on the classifier, the next subsection is dedicated to explain the details of such module.

---

**Algorithm 1:** Extraction of the main decorated surface $S$ given a fragment $F$ and a plane threshold $Th$.

---

$P$ = ExtractDominantPlanes($F,Th$);
$S$ = GetDecoratedSurface($P$);
**ExtractDominantPlanes** ($PCD, Th$)
    **inputs :** point-cloud $PCD$ of a factured fragment , RANSAC Plane Threshold $Th$
    **output:** List $P$ of dominant planes extracted from $PCD$
    **do**
        Extract dominant plane $P_i$ from $PCD$ through RANSAC;
    **while** $P_i$ *satisfies Th*;
    **return** $P$;

**GetDecoratedSurface** ($P$)
    **inputs :** List $P$ of dominant planes, **BackgroundColorClassifier**
    **output:** Decorated Surface
    **foreach** *point-cloud* $P_i \in P$ **do**
        Classify decoration colors through **BackgroundColorClassifier**;
        Estimate decoration percentage $dp$;
    **return** $P_i$ *with the highest dp*;

---

### 3.1.1. Training Data for the Background Color Classifier

The fragments' deterioration poses multiple difficulties for the design of the decoration color classifier. For instance, the prominent decoration colors cannot be directly mapped to a specific tone palette because fragment surface erosion might have modified their intensities. In addition, because of damage and vandalism that heritage assets have undergone over decades, some broken fragments might present scratches and graffiti which prevent from intuitively distinguishing relief decoration traces. The classification of decoration colors, as a consequence, unarguably requires human intervention. Therefore, we use machine learning to design the classifier. The implementation details of which are described in two parts, first an analyses of the training data is presented, after which the results of the obtained model are discussed.

Instead of considering the remaining paint of the decorated surface as inliers or regular observations, we use the colors of non-decorated surfaces to train the classifier. As these colors reassemble the decoration background, we assume that the outliers of the trained classifier potentially correspond to decoration colors. This is a convenient training approach since it solely requires us to identify the decorated surface, exclude this region, and leave the remaining point-cloud as training data. This process is done via a custom-made virtual platform that allows for smoothly maneuvering the fragments in 3D space to identify the decorated surface. Once the operator finds the decoration, he indicates four points across the decorated area; after which a region growing approach is conducted to extract the complete surface. This results in two point-clouds: The main decorated region and the rest. The former is used to create a ground-truth data-set to calculate the success rate of the proposed method, whereas the colors of the latter point-cloud are processed to generate the training data.

In order to prevent over-fitting when training the model, the input colors are clustered in $n$ prominent background tones by means of the K-means algorithm [15], a method that groups observations based on well-identified K centroids within the data-set. As such, this clustering process is performed not only to create the training observations but also as a prior step to classification. The training data is composed of the 6000 most prominent RGB colors obtained from a set of 600 fractured fragments. Figure 3 shows the histogram of the training data along with its pallet of prominent colors. The histogram's distribution indicates that it is feasible to model a machine-learning system with the acquired data since the range of RGB values can be delineated by a well-defined learning frontier. This property is better understood observing the pallet of colors (Figure 3 right), which shows that training samples are defined by the RGB-space of light brown tones. These observations are expected as the particular scanned fragments are made of limestone.

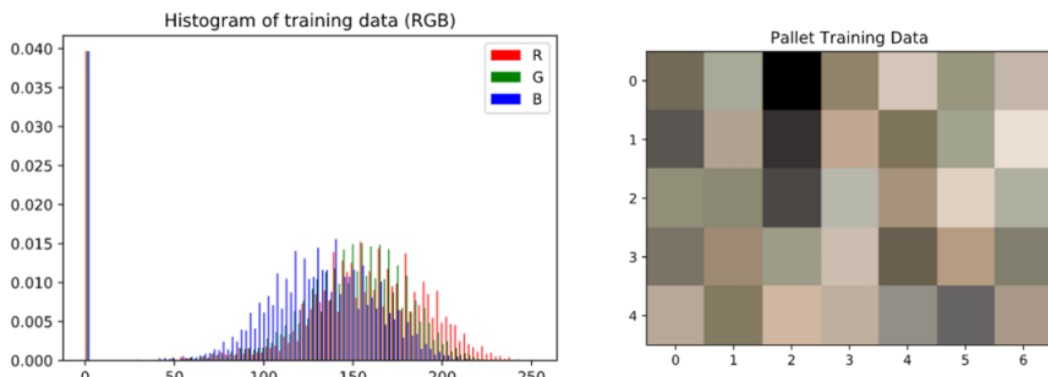

**Figure 3.** RGB color histrogram of training samples and the 35 most prominent colors of the distribution.

### 3.1.2. SVM Model

The training data is fed to a computational module that aims at identifying decoration by detecting observations whose properties differ from the training samples. Hence the nucleus of this module is an outlier detector. Multiple probabilistic techniques such as the Isolation Forest [16],

Local Outlier Factor [17], and One-Class Support Vector Machine (SVM) [18] have been proposed to detect abnormal data from a set of regular observations. In general, SVMs generate either a classification or a regression function that separates the observed data into multiple groups or classes through an N-dimensional hyperplane. One-Class SVM, more specifically, calculates an input-output mapping function, which estimates whether an observation is an outlier or inlier. This unsupervised approach is robust against irregular sampling distributions since it employs non-linear regression kernels to delineate the boundary of inliers. These characteristics make this model a suitable option to detect background colors.

Although at first glance the RGB-space of training samples seems to be defined by brown-like colors, their irregular distribution in the color space suggests modeling the One-Class SVM through a Radial Basis Function (RBF) kernel rather than a hyperplane. Figure 4 shows the obtained classification results in 2D space for two possible combinations: Green-red and blue-red. The training data is marked with white circles and the abnormal observations are depicted by yellow points. The latter are potential decorated colors and all points within the learned frontier correspond to background. Note that some irregular observations lie quite close to the decision boundary. This phenomenon is due to the gradual erosion of paint that turns vivid colors into pastel tones, affecting both decoration and background. Even though this issue makes the model prone to yield false positives, these errors do not influence the overall classification performance for the decorated surface extraction. The decoration/background classifier's output is solely used by Algorithm 1 to estimate the rough amount of decoration paint on an extracted plane and determine its likelihood of being the main decorated fragment's surface. Therefore, the misclassified points can be permitted since we are interested in computing an approximate percentage of paint on a surface rather than accurately delineating the decorated points.

Figure 5 shows some significant results of the proposed approach for the main decorated surface segmentation. The segmented point clouds were obtained by considering both color decoration and planarity. To clearly show the classifier's outcome, the segmented point-cloud and the k-means clustering for k = 10 are depicted. The background points are colored green and the outliers (corresponding to paint) are blue. Additionally, based on the obtained classifier, the resulting dominant colors are labeled as inliers for background and as outliers for potential decorated points. As expected, the majority of outliers, specially those tones that tend to red or blue, correspond to actual decoration. As mentioned, although there are misclassified points, these false positives do not influence the extraction results since the background classification is only used to estimate the percentage of decoration on the surface. As noted in the last fragment (bottom right), the classifier did not detect any decoration color. Therefore, planarity is considered to determine the main decorated surface.

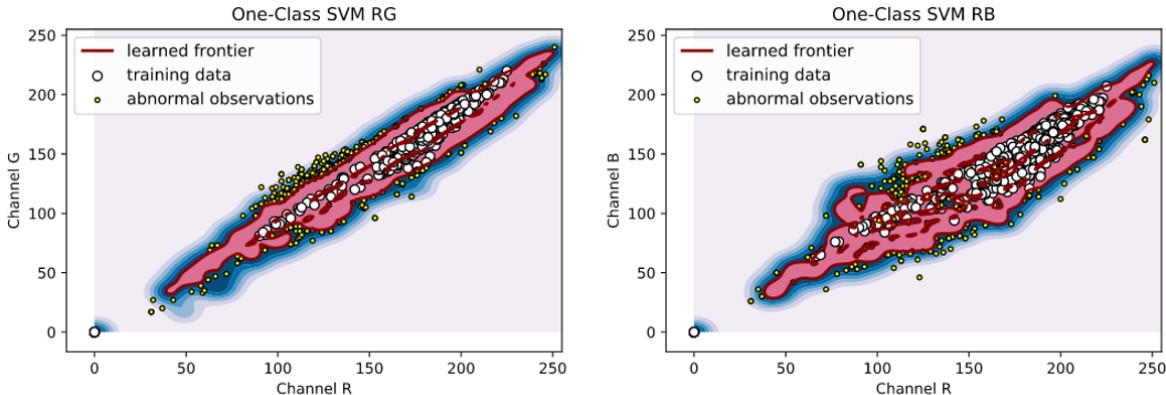

**Figure 4.** Resulting trained-frontier for the background color classifier considering two color channel combinations.

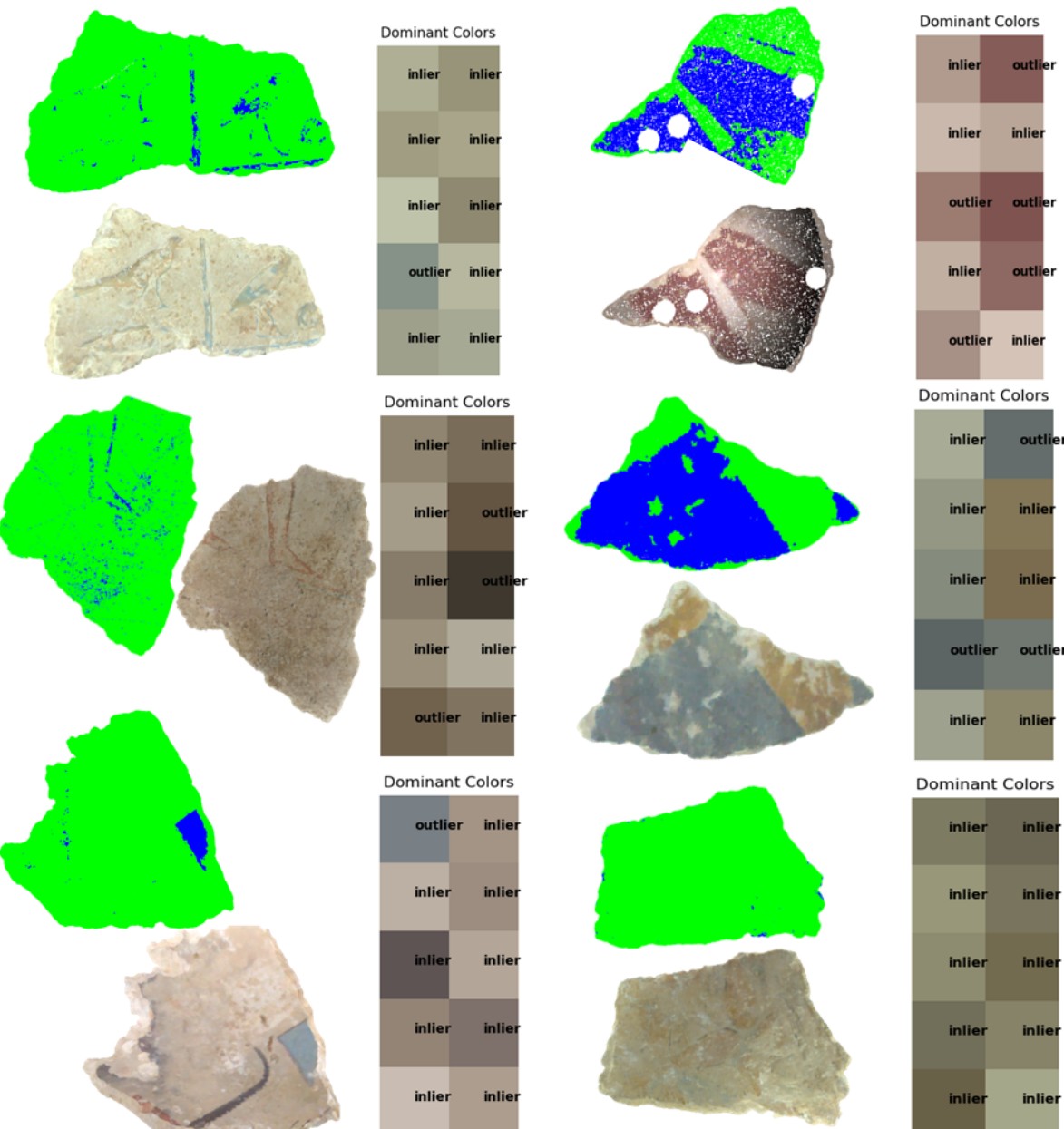

**Figure 5.** An example of main decorated surface extraction. The classified point-cloud depicts background colors marked in green and decoration colors marked in blue. Next to each point-cloud, the prominent colors of the surface are shown along with their label according to the classification result.

### 3.2. Automation

The previous section focused on the method to improve the segmentation of the main decorated plane, which serves as a baseline to extract a fragment's contour. This section describes the computational components integrated into the Puzzling Engine to increase its degree of automation. More specifically, Figure 6 shows the algorithmic modules of the registration pipeline that are implemented for the engine to be capable of estimating matches and propose alignment solutions. The input of this registration workflow is twofold: The segmented contour surface of both a query object and a target object. The first step consists of extracting salient points from the input data to describe their geometric properties. These descriptions are carried out locally, which means that for each region of interest a 3D feature descriptor is constructed. The obtained floating point descriptors are matched in Euclidean space, resulting in a dense set of putative correspondences. A novel method for outliers removal is introduced in the pipeline to reduce the number of false positive matches.

This technique uses geometric-based matching heuristics to filter out incoherent matches and is described in Section 3.2.2. The final set of estimated correspondences is deployed to calculate a 3D rigid body transformation between input surfaces, resulting in an alignment solution that can be improved by the user by means of the set of registration tools presented in our previous work.

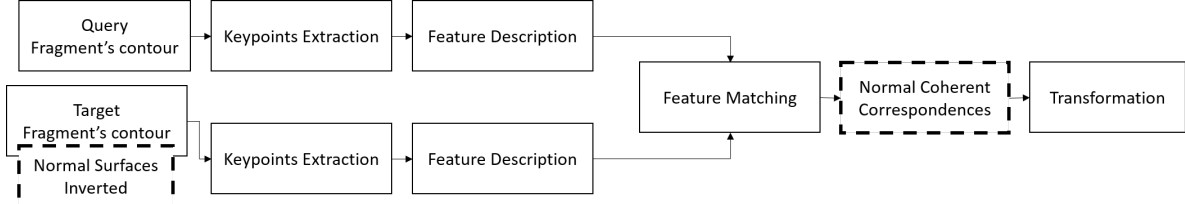

**Figure 6.** The pipeline for automatic registration of fractured fragments. The rectangles marked with dotted lines indicate the additional stages incorporated in the pipeline to cope with the geometrical constraints of heritage assets.

### 3.2.1. Feature Extraction and Description

3D feature-based methods aim to describe an object in terms of its geometric properties. Typically, this description is either based on the 3D object as a whole or on regions of interest, hence their classification as global and local features respectively. Both approaches have been applied to tracking systems, localization modules, object retrieval, automatic registration, etc. Due to the wide range of applications, these methods have been extensively studied over the last few years. The ideal descriptor exhibits a good trade-off between compactness, low-computational burden, robustness against nuisances, transformation-invariance, and distinctiveness. By taking these evaluation metrics into consideration, Han Xian-Feng et al. [19] presented a review of both global and local 3D descriptors and their performance on point-clouds for various applications. The assessment indicates that local descriptors are well-suited for 3D registration and matching applications. Unlike global-based approaches, local descriptors proved to compute accurate geometric descriptions of regions of interest, while also being robust against occlusions, noisy points, and data incompleteness. These characteristics make this approach viable to tackle the problem of automatic fragments reassembling.

Local features methods rely on two fundamental steps: Keypoints extraction and feature description. The former step aims to find salient points of interest, whose neighboring regions are geometrically distinctive. The latter step constructs descriptors by means of diverse computations such as local-transformations, normal histograms, curvatures, and so on. Guo Yulan et al. [20] conducted a detailed evaluation of the most used local features by exploring different combinations between keypoints extractors and descriptors considering not only the final application but also the acquisition technique of the input data. According to the obtained results, for dense point-clouds recorded with 3D professional scanners, the ISS3D [21] feature extractor combined with either of these descriptors: Rotational Projection Statistics (RoPS) [22], Fast Point Feature Histograms (FPFH) [23], or Unique Shape Context (USC) [24], are suitable approaches for matching.

The ISS3D algorithm extracts points of interest by calculating shape-based discriminative measurements from clusters of points defined across the point-cloud. As for 3D features, the FPFH algorithm is a discriminative descriptor that computes a histogram of features from a set of geometrical operations such as normal surface and curvature. Along the same lines, inspired by the 3DSC feature [25], the USC descriptor computes a floating point vector that describes local regions in terms of their shape while offering a good balance between memory footprint and matching accuracy. Unlike the aforementioned features, the RoPS descriptor computes transformation-based geometric properties of salient regions by performing statistical calculations on points lying on the mesh's surface. Therefore, this feature requires input not only from the mesh's vertices but also the triangles of the polygonal structure. As we are processing point clouds, the FPFH and USC descriptors are tested to describe the segmented contour surface of similar fragments in terms of their geometric properties.

Once the contour surface's features are extracted and described, the resulting floating arrays are matched following the Nearest Neighbor Search (NNS) approach [26]. To efficiently perform this process, the descriptors are arranged into a k-d tree structure, speeding up the search for similar floating-point vectors. This matching approach has been widely used to rapidly find similar numerical vectors from large datasets and applied to match features of both images and 3D surfaces. The first set of obtained correspondences is highly prone to false positives because the distance in feature space is not a strong matching criterion. This problem is due to the lack of overlap and similar geometry footprint between fractured surfaces. For example, if counterpart fragments match together, the normal surfaces of their joint regions point in opposite directions. This results in two negative outcomes: Numerous mismatches and nonsense alignments. In order to overcome these difficulties, we introduce normal-coherent-based matching heuristics in the registration pipeline. More specifically, as shown in Figure 6, these heuristics are applied to the target fragment's contour and to the set of putative correspondences prior to estimating the transformation for fragments alignment.

### 3.2.2. Normal Coherence Feature Matching

Inspired by the pairwise alignment heuristics introduced by Mellado Nicolas [4] to semi-automatically reassemble archaeological artifacts, the idea behind normal coherence is adapted in order for matching algorithms to cope with the typical issues when dealing with archaeological data. Normal Coherence is a registration metric that evaluates the likelihood of counterpart fragments matching by comparing the normal direction of a pair of points. As shown in Figure 7, the normal orientations of geometrically-related surfaces point at opposite directions. Although this property is fundamental to determine matching surfaces, their descriptors will not be similar because the normals vectors differ in orientation. Therefore, the normal vectors of target objects are inverted in order for potential matching regions to exhibit similar geometric properties. This way, the matching process between local descriptors is calculated from an unchanged query object against descriptors of a target point-cloud with altered normals. The feature matching step, however, might still yield normal incoherent correspondences due to the erratic shape of the input surfaces. To filter out false positives, the normal orientation of matching points is checked when performing RANSAC, so that the estimated set of correspondences meet the normal coherence matching conditions. This approach was already deployed to fit planes to indoors scenes by Qian Xiangfei et al. [27]. In which sampling points with different orientation properties are discarded prior to fitting the input data to the plane equation. In our case, non-coherent points are removed in each RANSAC iteration before estimating the optimal rigid transformation.

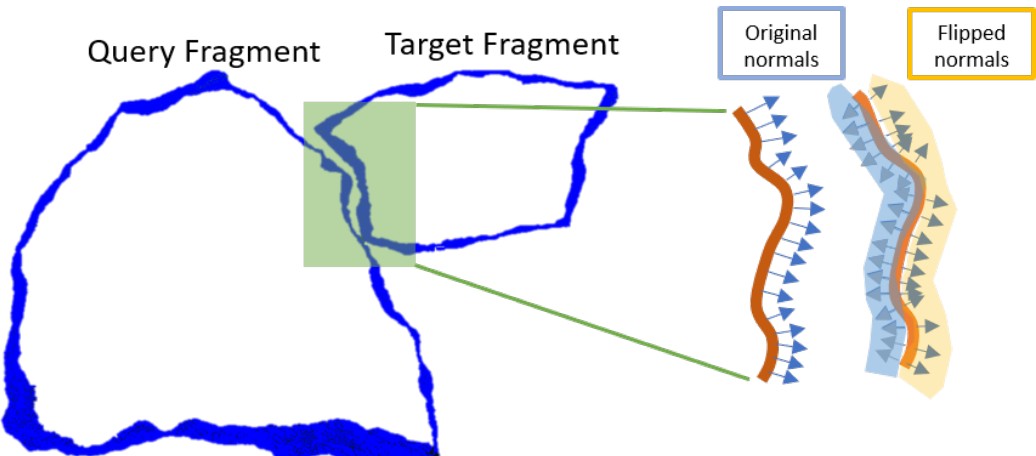

**Figure 7.** The normal coherence constraint. The target contour's normal surfaces are flipped to meet the normal-coherence criterion and obtain similar descriptors.

The last stage of the automatic registration workflow consists of translating and rotating the points of the query fragment according to the transformation estimated by RANSAC. Thus the query fragment is roughly aligned with respect to the target fragment. This alignment might serve as a starting point for the user to puzzle the input fragments. Once the user estimates an alignment through the virtual interface, a fine registration is automatically performed through the engine by deploying ICP on the closest points of counterpart fragments. This process only takes into account alignments along $x, y$ axes and roll angle since fragments are already aligned with respect to the $z$ axis.

## 4. Evaluation of Registration Pipeline

The previous section focused on the automatic registration algorithms of the Puzzling Engine. The aim of this section is to evaluate the fragments alignment workflow. To this end, we employ a large dataset of ancient Egyptian fragments, which were excavated over the last decade. Due to field research time limitations and the sheer quantity of archaeological assets, it is unfeasible to find similar fragments and puzzle them on site. Moreover, their high degree of deterioration hampers experts to easily identify intuitive joints. Since fragments are not clustered according to distinctive characteristics such as color, shape, size, excavation place, etc., the proposed pipeline is used to perform a massive search for potential matching possibilities. The results of this brute force matching are evaluated by experts to determine which of them are actual fits. After discarding false positives, the correct matching solutions are manually aligned via the puzzling engine tools. The final accurate alignments approved by experts serve as ground truth metrics to compare the proposed Normal-Coherence-based matching approach against similar featured-based registration methods. The evaluation criteria include a comparison of computed 3D-rigid body motions with ground truth transformations, processing time of matching algorithms, and frame rendering of the virtual platform.

### 4.1. Dataset of Fractured Fragments

Our data-set is composed of a large group of broken decorated wall fragments that were found in the elite cemetery of Dayr al-Bersha, Egypt. The fragments originally belonged to the colossal decorated tombs that date back to the Middle Kingdom period of ancient Egypt, from 1975 BC to 1640 BC. The current ruinous state of the site, due to natural catastrophes, looting activities, and historical events, led researchers of KU Leuven University to preserve and further study the site. During their on-site missions, they excavated thousands of decorated stone blocks of different sizes and characteristics. On average, the size ranges from 10 cm to 50 cm long, are made of limestone, and in most of cases contain a flat surface with decoration. This is a region of interest for experts since it contains remaining traces of either inscriptions or drawings that are identified through decoration colors or relief-based carvings. The main characteristic of this collection, for the current research purpose and according to experts, is that the likelihood of finding some fragments matching together is high. However, due to the great amount of fragments, this is a laborious task.

The digital acquisition of ancient archaeological artifacts poses multiple challenges. For the wall-decorated fragments, in particular, it is primordial to be able to capture every surface of the object while preserving their archaeological properties. Additionally, in order for the registration algorithms to yield reliable results, the input data must satisfy accuracy requirements in terms of homogeneous brightness, scale, and geometric information. Hence structured light technology is deployed to produce accurate dense models of the fragments. More specifically, the 3D scanner *Einscan Pro Plus* [28] is used for acquisition since it is equipped with an external camera and two flashlights to capture color under controlled light conditions. Furthermore, it is capable of recording point-clouds at a 0.2–3 mm point distance resolution. Finally, the raw 3D data is processed to generate 1500 polygon mesh models, whose number of faces varies from 0.4 to 1.4 M.

*4.2. Brute-Force-Based Approach for Ground Truth Metrics*

To obtain ground truth registration metrics from the digitized fragments, a brute force matching is conducted on the basis of the proposed registration pipeline, combined with the manual alignment tools of the engine. First, the contour point-clouds of the main decorated surfaces, which were extracted when generating the color classifier's training data, serve as input for the proposed registration workflow. The ISS key-point extractor is combined with FPFH and USC descriptors, complying with the Normal Coherence requirements. To err on the side of caution, both the feature extractor and descriptors were computed for different search radii. Then, a brute force matching is performed for all computed descriptors not only to estimate correspondences but also to calculate a rigid-body transformation for automatic alignment. This results in a vast group of potential pairwise puzzles. Consequently, the second step consisted of filtering out erroneous solutions and improving the estimated rigid transformations. The latter is done by experts via the registration tools of the puzzling engine. For sake of computing consistent ground truth metrics, the puzzling engine is slightly modified to allow the user to solve the puzzle by only maneuvering the query fragment. Once a fine alignment is manually determined, the engine computes the rigid body motion that transforms the query fragment's initial position to the manually modified position. This process is done by means of Singular Value Decomposition (SVD), a factorization method that computes the optimal rotation and translation between a group of transformed points. The final set of obtained transformation matrices serves as ground truth metrics.

The aforementioned steps are computationally expensive since they entail time-consuming mathematical operations. We are dealing with hundreds of mesh models and the computational complexity of the brute force matching process itself is $\mathcal{O}(n^2)$, where $n$ is the number of fragments. Therefore, this problem is addressed through ubiquitous computing using HTCondor [29], an open-source software solution that allows for performing intensive tasks while fairly distributing the computational burden over multiple computational cores. Figure 8 shows an overview of the computational processes that are managed by HTCondor. Prior to brute force matching, feature extraction and description processes are submitted to HTCondor so that these tasks are performed in parallel for every single fragment. The output of such processes consisted of a list of keypoints and descriptors encoded into binary files for each fragment. Once descriptors are computed, HTCondor manages the pairwise brute force matching for all possible fragment combinations and descriptor configurations (FPFH and USC with different search radii). The brute force matching process yielded 60 pairwise solutions in total. After these results were revised and improved by experts, the final group of actual puzzles consisted of 12 solutions. Thus, the precision of the brute force approach is 0.2. The recall cannot be determined since we do not know the actual number of pairwise solutions. The final group of pairwise solutions determined by experts is illustrated in Figure 9. Although this nondiscriminatory matching process is certainly prone to yield numerous false positives, it helped determine ground truth metrics from a dataset where it was uncertain to find puzzles.

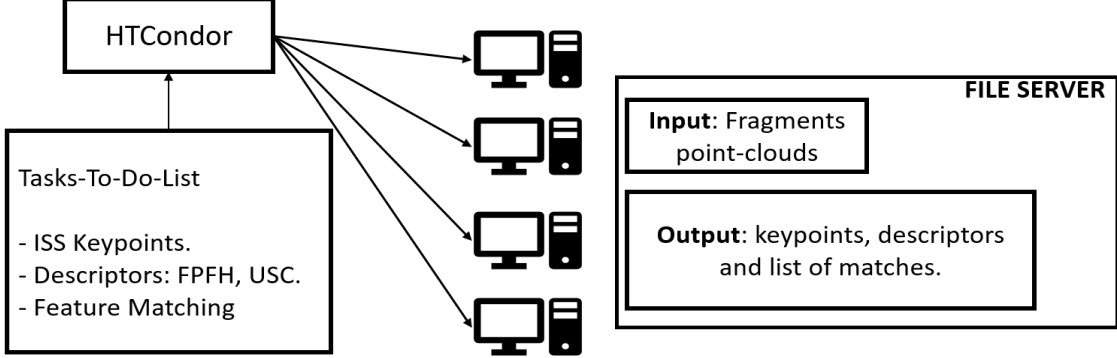

**Figure 8.** Jobs distribution in HTCondor to perform brute force matching.

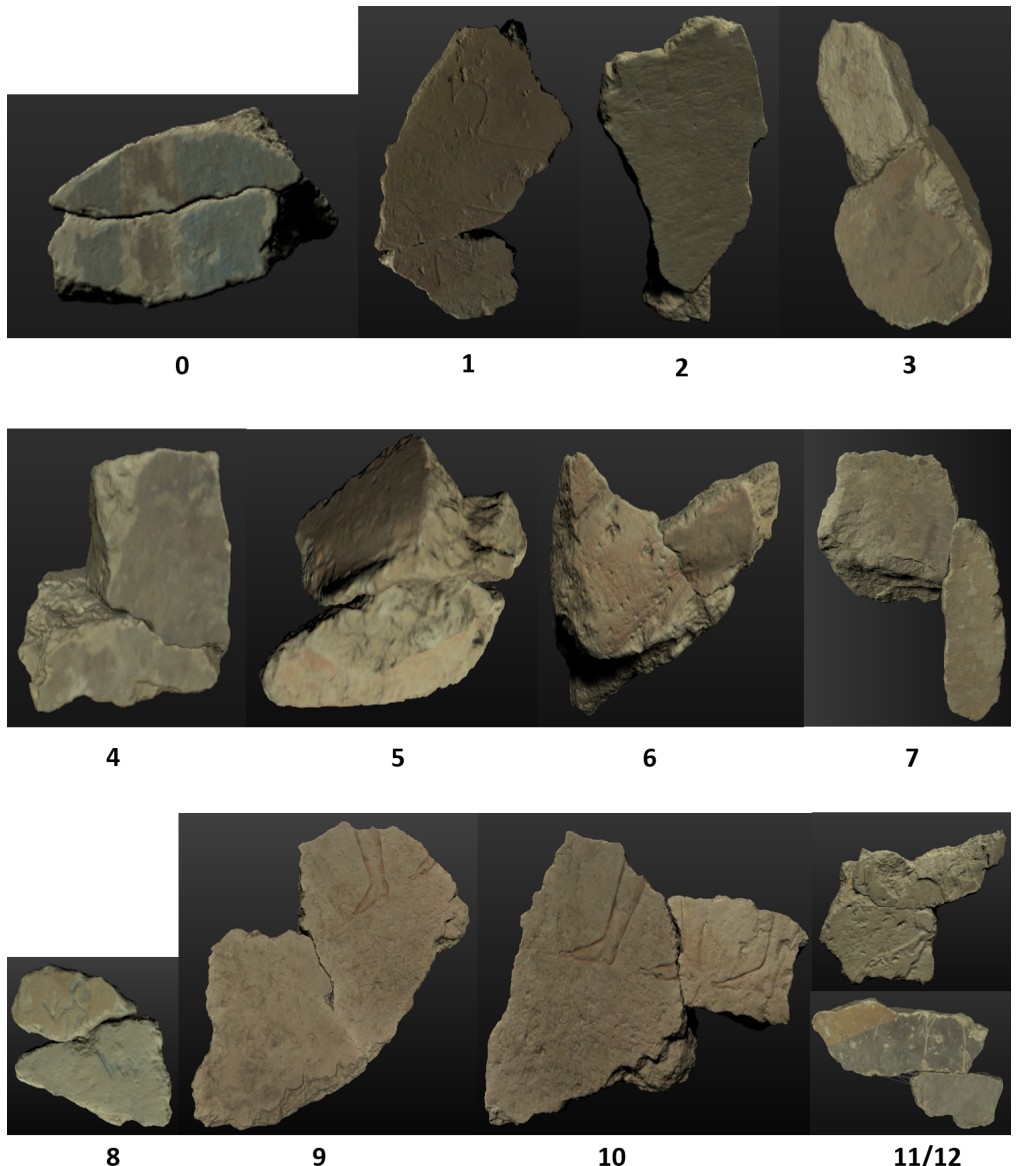

**Figure 9.** Brute force matching results: Pairwise solutions resulting from brute force matching approach. The matches were revised by experts who also refined the fragments' alignment through the puzzling engine. The obtained rigid transformation are used as ground truth metrics to evaluate the proposed registration modules.

### 4.3. Accuracy of Normal Coherence-Based Registration Methods

We set up an experiment which seeks to evaluate the accuracy of 3D rigid body transformations estimated by the Normal Coherence-based registration algorithms. The ground truth transformation matrices estimated in the previous step are compared to a set of arrangements of features and matching methods. The experimental setup results serve as indicators to assess which algorithms combinations are suitable to increase the degree of automation of the puzzling engine. These include: *FPFH + normal coherence rejector*, *FPFH + distance threshold rejector*, *USC + normal coherence rejector*, and *USC + distance threshold rejector*. The accuracy comparisons are conducted by examining the error statistics proposed by Eggert D.W. et al. [30]. These metrics include the norm of difference between estimated translation vectors, $\|\hat{\mathbf{T}}_{alg} - \hat{\mathbf{T}}_{true}\|$ and the norm difference of unit quaternions representing the rotation, $\|\hat{\mathbf{q}}_{alg} - \hat{\mathbf{q}}_{true}\|$. Finally, the obtained reference transformation matrix and the one estimated from the set of correspondences are applied to the query fragment's point-cloud. The Root Mean Square (RMS)

error of the distance between such points is calculated by Equation (1), where $N$ is the number of points and $\mathbf{PCD}_{alg}$ and $\mathbf{PCD}_{true}$ are the transformed point-clouds of registration algorithms and ground truth data respectively:

$$RMS_{error} = \sqrt{\frac{\sum\limits_{}^{N}\|\mathbf{PCD}_{alg} - \mathbf{PCD}_{true}\|^2}{N - 3}}. \tag{1}$$

The goal of this registration metric is to measure the accuracy of the computed transformation matrices' elements. The metric comparisons are carried out in Euclidean space for both the 3-dimensional translation vector and the 4-element vector of orientation quaternions extracted from the $3 \times 3$ rotation sub-matrix. Moreover, the RMS error of automatic alignments is calculated to measure the overall influence of the estimated transformation when applying it by the query fragments' points.

Figure 10 shows the results for every pairwise solution. Each puzzle is listed according to its corresponding reference number indicated in Figure 9. The errors are represented in units of the puzzling engine's global coordinate system, which are given in centimeters since the recorded models were scaled from millimeters to centimeters. As noted in Figure 10a,b, translation and rotation plots, for most of the solutions the combination of FPFH and NC yields better results since the error difference is close to zero. When NC is considered for the FPFH descriptor, the results are highly improved. For instance, for solutions 0, 2, 8, and 11, the translation error difference between FPFH and FPFH NC is larger than six units. The same behavior can be observed in the rotation plot, but in this particular case the errors are much smaller. Rotation and translation errors provide an insight into the variance of the estimated rigid body transformations' parameters. However they do not provide us with information on the registration accuracy.

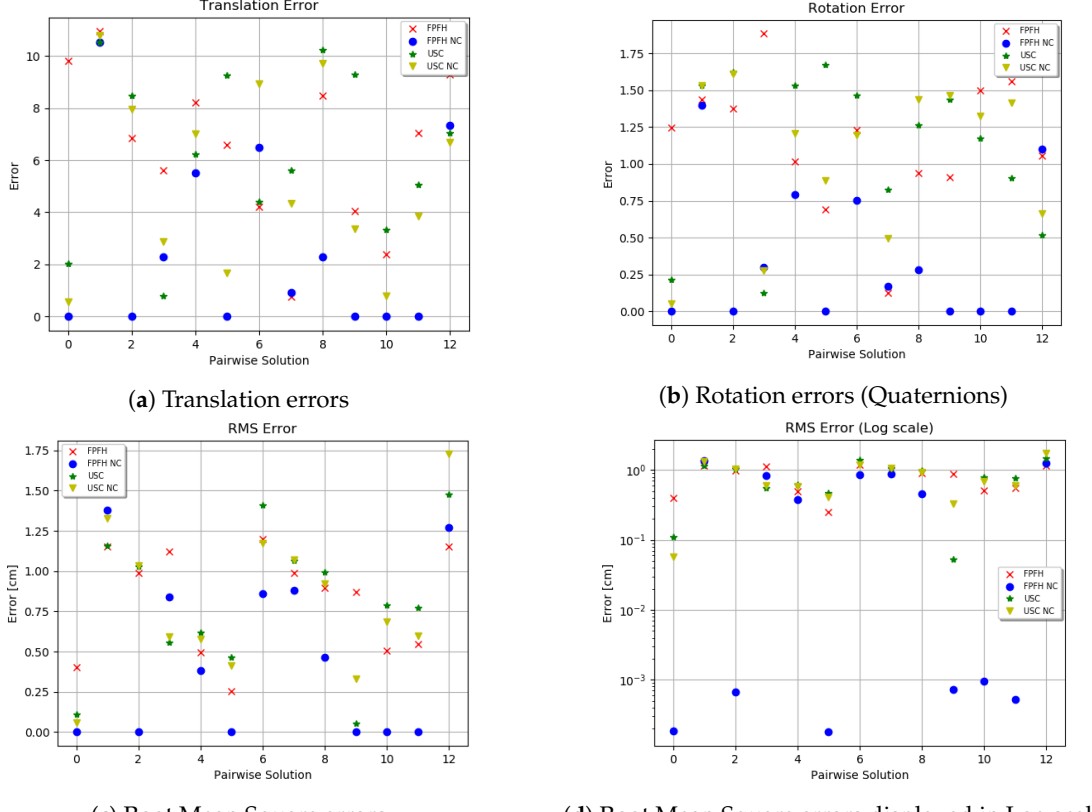

(**a**) Translation errors

(**b**) Rotation errors (Quaternions)

(**c**) Root Mean Square errors

(**d**) Root Mean Square errors displayed in Log scale

**Figure 10.** Evaluation of 3D rigid body transformations computed by matching USC and FPFH local descriptors based on multiple statistic errors. Additionally, the normal-based matching heuristics are included to highlight the positive impact of the proposed modules.

The RMS error, on the other hand, measures the registration performance or the difference in distance between aligned models. This metric evaluates the rigid body transformation when applied to the query fragment. Hence a close-to-zero error means that the automatic alignment is similar to the ground truth. As expected, the FPFH feature with the NC matching approach yielded better results since errors fluctuated between 0 and $10^{-3}$ for 50% of the found solutions. This means that only slight manual alignments are necessary to accurately register the fragments. To illustrate registration errors, Figure 11 depicts the pairwise solutions zero, six, and 12. These correspond to the combinations FPFH + NC, USC + distance threshold rejector and USC + NC, whose errors are 0.00018, 1.40775, and 1.72560 respectively. As noted in Figure 11a, when the error is very low, there is an imperceptible difference with the ground truth alignment. However, when errors are larger, misalignments start to become much more visible. For example, Figure 11b shows an incorrect alignment since the query object is penetrating the target fragment's surfaces (see orange line). In Figure 11c, although there is no overlap between fragments and at first glance the alignment seems correct, when comparing the result to the ground truth, the query fragment's rotation and translation are erroneous.

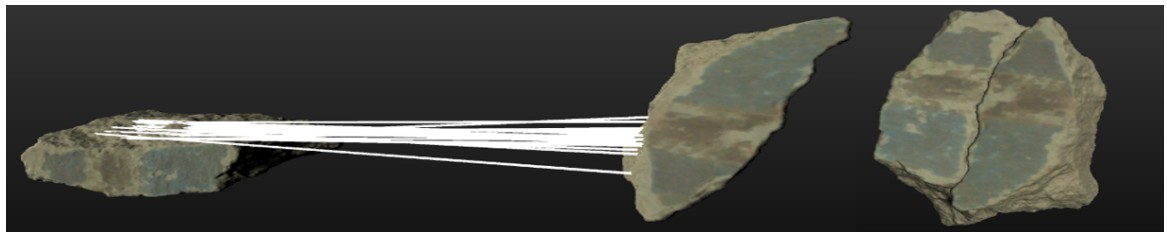

**(a)** Solution №0, *FPFH + NC-based matching*

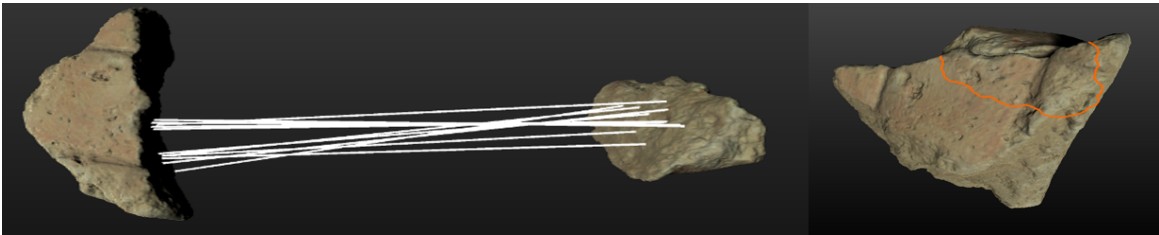

**(b)** Solution №6, *USC + Threshold-based matching*

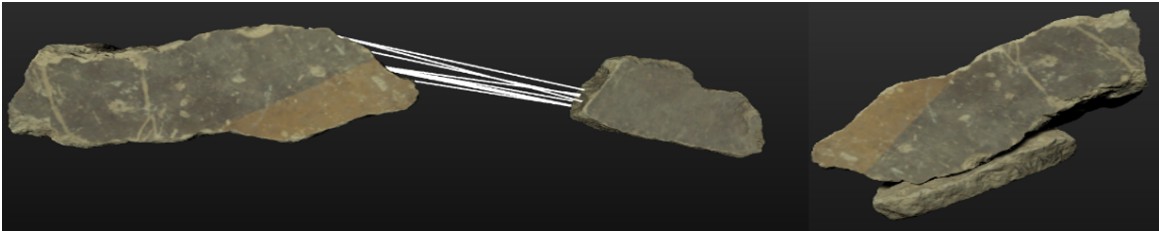

**(c)** Solution №12, *USC + NC-based matching*

**Figure 11.** Examples of automatic matching results. On the left side, the computed correspondence points are visualized though matching lines. On the right side, the estimated rigid body transformation.

This experimental framework shows that the proposed registration pipeline is well-suited to reassemble heritage fragments. In spite of the fact that this approach might estimate misalignments, the matching lines proved to serve as a visual guide to rapidly identify potential matching regions. Moreover, the normal coherence approach highly improves the registration performance, so that for some fragments only minor alignment corrections are needed. When this approach is combined with the FPFH feature the registration error is close to zero. Therefore, these algorithms are integrated into the game engine to complement the manual registration tools.

Time-Processing Metrics

Real-Time performance is an important evaluation metric when developing a User Interface. It indicates how fast the engine's algorithms respond to user operations under different types of input data. In addition, it provides insights into the algorithms' efficiency in terms of time-processing and memory footprint. Time complexity and frames per second (fps) are two metrics that provide fruitful information to evaluate such a performance. The puzzling engine computes two tasks on the fly: 3D rigid body maneuvers and pairwise matching. Therefore, we estimate both the explored feature correspondence algorithms' time complexity as well as the engine's fps. The experiments are performed on the following system: Intel Core i7-7700 at 2.6GHz, 16GB RAM memory, graphic card Nvidia GTX1050, 1493MHz GPU clock, Dell, Ghent, Belgium.

Table 1 shows the processing time of the feature matching algorithms. The first two columns correspond to the point-cloud density of query and target models. The other columns indicate the deployed feature descriptors along with their array length. The processing time is given in milliseconds and only measures the matching process. This constitutes arranging the descriptors arrays into k-d trees, removing false positives and euclidean distance-based comparisons. Note that matching FPFH features is far faster than USC due to their compactness: The former are constructed of a 33-elements array, whereas the latter descriptor contains 1960 elements. The time complexity of the NC-based matching is higher since it entails an extra iterative step to discard non-Normal-Coherent correspondences. As for the engine's rendering, Figure 12 illustrates the fps over a period of time, during which multiple tasks are performed, such as fragments maneuvering, loading multiple mesh models, rendering matching lines, registration, and so on. The fps is quite stable since the average fps is 40.79 and a standard deviation of 1.69 over 60 s. The obtained results satisfy the 30 Hz real-time rate requirement, which guarantees a smooth interaction between user and 3D models.

**Table 1.** Processing time that takes to compute the registration pipeline of Figure 6.

| # Query Points | # Target Points | USC NC 1960 [ms] | USC 1960 [ms] | FPFH NC 33 [ms] | FPFH 33 [ms] |
|---|---|---|---|---|---|
| 1322 | 1254 | 8239 | 6554 | 89 | 87 |
| 1322 | 748 | 5546 | 4229 | 115 | 81 |
| 1465 | 4331 | 21,500 | 21,485 | 246 | 221 |
| 1490 | 2086 | 11,684 | 11,173 | 154 | 120 |
| 1708 | 4458 | 29,442 | 24,982 | 243 | 215 |
| 2085 | 5010 | 34,774 | 34,149 | 278 | 268 |
| 2091 | 1394 | 12,328 | 10,343 | 163 | 109 |
| 2414 | 2085 | 19,021 | 17,066 | 189 | 177 |
| 2653 | 3550 | 31,065 | 30,989 | 262 | 239 |
| 3273 | 5925 | 67,855 | 58,231 | 332 | 299 |
| 3324 | 2931 | 32,272 | 30,229 | 276 | 234 |
| 4116 | 2026 | 30,931 | 30,035 | 242 | 219 |
| 5272 | 1925 | 33,128 | 34,302 | 261 | 257 |
| 6228 | 3072 | 57,280 | 56,295 | 369 | 352 |
| 6745 | 5925 | 132,375 | 114,226 | 492 | 434 |

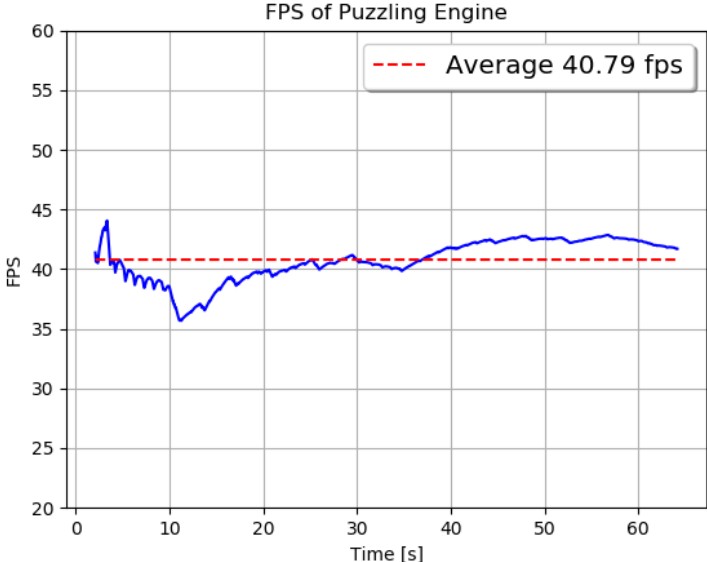

**Figure 12.** Rendering performance of the Puzzling Engine over a period of time. Actions like loading fragments, displaying matches, and applying transformations are carried out during this interval.

## 5. Conclusions

In this paper a novel Puzzling Engine was proposed for ancient wall-decorated fragments that takes elements of both kinds of reassembling approaches: User-assisted and fully automatic. On the one hand, the puzzling process is performed through a virtual interface that facilitates the 3D manipulation of objects as well as segmenting high level properties of the fragments to intuitively discern matching regions. On the other hand, it takes the key approach of automatic registration pipelines in order to increase its degree of automation when the user is looking for matching regions for alignment. The experimental framework consists of finding pairwise puzzle solutions from a set of more than one thousand wall decorated blocks made of limestone. As proven in the experiments, this puzzling engine was capable of matching fractured stones even in cases where erosion had highly deteriorated the decoration or the matching surface. Although for some fragment samples the registration pipeline alone was capable of computing accurate alignment transformations without user interaction, this approach does not intend to compete against automatic reassembling approaches. The Puzzling Engine aims to support restorers in the matching labor by means of comprehensive and sophisticated registration tools.

As a second key result, this paper presents a study of widely-used 3D features in the context of fragments reassembling. Even though these features have been broadly deployed for object retrieval and robot navigation applications, the proposed work proves that they can be applied to digital heritage reassembling as well. However, the typical feature-based registration pipeline by itself does not provide accurate correspondences. Therefore, we propose to incorporate Normal-Coherent-based matching heuristics to the pipeline in order to obtain reliable matches. The obtained results are quite favorable and provide insights into possible ways to automatically match heavily deteriorated fractured surfaces. Although the puzzling engine requires human expertise to yield accurate results, manual assistance is needed only in the last stage of the registration workflow: Matching validation and fine alignment. Hence future work aims at increasing the degree of automation by a merge between local and global approaches. For instance, shape-based matching of boundary contours for rough alignment and local geometric features for fine registration. This approach might be complemented by considering additional matching cues such as decoration content, or curves continuity of hieroglyphics between counterpart fragments. Additionally, a culling step to filter out incompatible assets, prior to performing the registration pipeline, would reduce the computational burden of the brute force searching. This can

be done by considering the scale, color, and global shape of the fragment before computing more complex geometry-based matching operations. These additional matching characteristics can be applied to puzzle not only decorated stone fragments but also other types of 3D heritage artifacts such as broken pottery, ancient figurines, sculptures, etc.

**Author Contributions:** R.d.L.-H. and M.V. contributed equally to the work. All authors have read and agreed to the published version of the manuscript.

**Funding:** The research presented here features within the *Puzzling Tombs* project (nr. 3H170337), funded by the KU Leuven *Bijzonder Onderzoeksfonds*.

**Acknowledgments:** We thank Suzanna Cuypers for her support in the experimental framework.

**Conflicts of Interest:** The authors declare no conflict of interest.

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
