# Peer review of "A Hybrid Approach to Reassemble Ancient Decorated Block Fragments through a 3D Puzzling Engine"

_remotesensing, doi:10.3390/rs12162526_

Round 1

Reviewer 1 Report

The topic about the 3D puzzling engine to reassemble ancient heritage fragments is full of significance in heritage conservation. It is a new method proposed in recent years. However, previous studies about this topic is focused and deep. It is not clear whether this manuscript do a remarkable update to previous method and where is it? I am so sorry I choose to reject it due to the reasons above. 

Author Response

Thank you for the review. Indeed, as mentioned in the related work, multiple studies propose approaches to reassemble fragments either automatically or manually (via a virtual environment). In the same section, from line 122 to 126, we point out the main downside of current automatic approaches when dealing with actual heritage fragments. Motivated by this constraint, we propose to combine both techniques, manual and automatic, to facilitate the puzzling of highly damaged broken fragments. This combination of methods is one of the main differences with the state of the art. Along the same lines, another major difference is that the proposed automatic registration pipeline is tested to find puzzles of real ancient Egyptian fragmented blocks, instead of generic broken artifacts. Additionally, to emphasize the algorithmic differences w.r.t. the state of the art, from line 59 to 66, we outline the key contributions of the paper.

Reviewer 2 Report

This is a clearly written paper on automatically reassemble ancient heritage fragments. The paper is  interesting and readers will find it useful. I read it carefully and I did not see anything that should be changed, just a very small writing mistake in Line 37 “paper. In which”. The manuscript structure, information flow, presentation, figures and tables look very good. I do not see any major changes to make. 

Author Response

Thanks for the review. We have rephrased the sentence in line 37.

Reviewer 3 Report

I am not a specialist in this kind of work but am excited about the possibilities it would open.  My only recommendation is to consider defining/explaining some of the vocabulary to make it more readable to archaeologists and curators who are less technically savvy.

Author Response

Thanks for the review. Indeed, in the methodology, some concepts were not clearly defined. As suggested, we have briefly explained some of the methods we are deploying:

Line 183, the K-means clustering algorithm is defined.

Line 285, the NN search, and K-d tree methods are explained in the context of feature matching.

Line 375, The SVD method is explained

Reviewer 4 Report

Good work.

Interesting and useful idea.

Clear and detailed methodology. 

detailed presentation of applied technologies.

It is recommended not to neglect the sector literature for the recomposition of the models.

It is hoped that the techniques used will also be applied to ancient fragments of other cultures besides the Egyptian one.

Author Response

Thanks for the review. Certainly, this approach aims to puzzle Egyptian fragments. However, as future work, it is considered to modify some computational elements of the automatic registration pipeline, so it can be applied to other kinds of artifacts like broken pottery, ancient figurines, sculptures, etc. This has been included in the last paragraph of the conclusions.

Reviewer 5 Report

There are indeed many - too many - repetitions in all sections of the article. This does not help the scientific nature of the article that should not to get lost in unnecessarily long exposures of already established concepts. I recommend a careful rereading by smoothing out all the unnecessary repetitions. I suggest instead giving more space to a less technical but more explanatory description of the virtual interface - to which in the article a specific section has not been dedicated - believing that this wold benefit the article as it would make it more accessible for understanding also for less experienced researchers in algorithmic issues.

Author Response

Thanks for the review. Some concepts are repeated in the introduction and section 3.2 because of twofold. On the one hand, a description and a figure explaining the proposed approach were included in the introduction to point out the differences with the conference paper [2]. On the other hand, a pipeline of the automation process is briefly described in the introductory paragraph of subsection 3.2 and Figure 6 since this section includes the key contributions of the paper. The details of each computational component are explained in each subsubsection.
As for the user interface, its description is not included in this paper since the technical details are published in the conference paper. This is mentioned in lines 37 and 38.

Reviewer 6 Report

The paper is well written and the process well presented. The idea is interesting and surely usefull. There are some typos and a moderate chack of English is required.

pg 7 line 210 and following: maybe a more clear explaination on why the issue presented does not influence the process is needed 

pg 10 line 309 and following maybe a more deepen explaination on how the query works is needed

Author Response

Thanks for the review.

The decoration/background classifier ’s output is solely used by Algorithm 1 to estimate the rough amount of decoration paint on an extracted plane and determine its likelihood of being the main decorated fragment’s surface. Therefore, the misclassified points can be permitted since we are interested in computing an approximate percentage of paint on a surface rather than accurately delineating the decorated points. This text has been added on page 7 line 215.

The last stage of the automatic registration workflow consists of translating and rotating the points of the query fragment according to the transformation estimated by RANSAC. Thus the query fragment is roughly aligned with respect to the target fragment. This text has been added on page 11 line 314.

Reviewer 7 Report

I really enjoyed reading your article. The paper is perfectly written, the scientific explanation is great and the most important thing, in my opinion, is the real applicability of your study as a support to the archaeologist. 

The only reason I've flagged the article for minor revisions is that references inside the text are not cited in order of appearance. 

Author Response

Thanks for the review. The references have been re-ordered.

Round 2

Reviewer 1 Report

Thanks for your explanation. I get your idea more clearly. However, in order to strengthen your innovation and value, I suggest revise your title, abstract and conclusion more clearly and specific. For example, your method which combined computer and human analysis and your research object should be emphasized. When I read your manuscript for the first time, I find some other articles such as “A hybrid human–computer approach for recovering incomplete cultural heritage pieces”, “3D puzzle reconstruction for archeological fragments”, “Digitizing Conservation: incorporating digital technologies for the reconstruction and loss compensation of archaeological ceramics”, “From Reassembly to Object Completion: A Complete Systems Pipeline”. They are similar study to some extent, but maybe different research object or updating method. So, It is need to emphasize your own special points different from previous study more clearly.

Author Response

Thank you for the feedback. Indeed, to some extent, the general aim of the state of the art is to propose a method to puzzle heritage fragments. However, since there is neither an automatic nor a user-assisted system capable of coping with all types of archaeological artifacts: pottery, sculptures, fragmented stones, etc., the input data is restricted to objects with similar features. In this case, we focus on a solution to puzzle real heritage wall decorated block fragments. Therefore, unlike the literature you mentioned, our registration strategy relies on characteristics such as planarity, fragments contour, and geometry-based properties. Following your suggestions, we have done the following modifications to the paper.

  1. The title was changed to "A hybrid approach to reassemble ancient decorated block fragments through a 3D puzzling engine", to emphasize our target objects.  
  2. We added a couple of sentences in the abstract (line 10) to differentiate our work to similar hybrid puzzling engines. " Unlike similar hybrid human-computer puzzling engines, our approach is capable of automatically proposing matches and rough alignments solely based on the fractured surfaces’ geometry. Based on these initial solutions and a set of registration tools, experts can accurately solve the puzzle."
  3. Finally, we added another sentence in the conclusion (line 482) to point out that the puzzling engine requires minimal assistance from the user, which is another key difference with similar approaches.  "Although the puzzling engine requires human expertise to yield accurate results, manual assistance is needed only in the last stage of the registration workflow: matching validation and fine alignment."